# Synthetic Notch-Receptor-Mediated Transmission of a Transient Signal into Permanent Information via CRISPR/Cas9-Based Genome Editing

**DOI:** 10.3390/cells9091929

**Published:** 2020-08-20

**Authors:** Malte Sgodda, Susanne Alfken, Axel Schambach, Reto Eggenschwiler, Pawel Fidzinski, Michael Hummel, Tobias Cantz

**Affiliations:** 1Translational Hepatology and Stem Cell Biology, Department of Gastroenterology, Hepatology and Endocrinology, REBIRTH-Center for Translational Regenerative Medicine, Hannover Medical School, 30625 Hannover, Germany; Alfken.Susanne@mh-hannover.de (S.A.); Eggenschwiler.Reto@mh-hannover.de (R.E.); 2Institute of Experimental Hematology, Hannover Medical School, 30625 Hannover, Germany; Schambach.Axel@mh-hannover.de; 3Division of Hematology/Oncology, Boston Children’s Hospital, Harvard Medical School, Boston, MA 02115, USA; 4Clinical and Experimental Epileptology, Department of Neurology, Charité—University Medicine Berlin, corporate member of Freie Universität Berlin and Humboldt-Universität zu Berlin and Berlin Institute of Health, NeuroCure Cluster of Excellence, 10117 Berlin, Germany; pawel.fidzinski@charite.de; 5Experimental Haematopathology, Institute for Pathology, Charité—University Medicine Berlin, corporate member of Freie Universität Berlin and Humboldt-Universität zu Berlin and Berlin Institute of Health, 10117 Berlin, Germany; michael.hummel@charite.de

**Keywords:** chimeric antigen receptor, CRISPR/Cas9-mediated gene editing, regulator of transcription activation (tTA), signal transformation, synNotch receptor signaling

## Abstract

Synthetic receptor biology and genome editing are emerging techniques, both of which are currently beginning to be used in preclinical and clinical applications. We were interested in whether a combination of these techniques approaches would allow for the generation of a novel type of reporter cell that would recognize transient cellular events through specifically designed synthetic receptors and would permanently store information about these events via associated gene editing. Reporting cells could be used in the future to detect alterations in the cellular microenvironment, including degenerative processes or malignant transformation into cancer cells. Here, we explored synthetic Notch (synNotch) receptors expressed in human embryonic kidney cells to investigate the efficacy of antigen recognition events in a time- and dose-dependent manner. First, we evaluated the most suitable conditions for synNotch expression based on dsRed-Express fluorophore expression. Then, we used a synNotch receptor coupled to transcriptional activators to induce the expression of a Cas9 nuclease targeted to a specific genomic DNA site. Our data demonstrate that recognition of various specific antigens via synNotch receptors robustly induced Cas9 expression and resulted in an indel formation frequency of 34.5%–45.5% at the targeted CXCR4 locus. These results provide proof of concept that reporter cells can be designed to recognize a given event and to store transient information permanently in their genomes.

## 1. Introduction

Intercellular signaling, one of the key mediators of organismic communication, is realized through the fine-tuned interplay of various molecular switches and structures. Unlike autocrine and paracrine signaling, juxtracrine signaling affects adjacent cells via cell membrane-bound ligands that target specific receptors. One prominent example of juxtracrine signaling in developmental biology is the notch receptor system. The binding of the delta ligand to the notch receptor induces cleavage of the intracellular receptor domain, which directly activates target gene transcription [1]. In essence, notch receptors have the ability to convert an antigen recognition signal into transcriptional activation [2], and the impact of synthetic Notch (synNotch) receptors, which introduce various bioengineered properties into that signaling system, is increasingly being recognized. For example, receptor cleavage-mediated transcriptional activation can be used for direct tracking of synNotch receptor-activated cells by fluorophore expression [3,4]. In another study, the synNotch receptor activation was used to trigger Interleukin 12 (IL-12) expression to recruit chimeric antigen receptor (CAR) expressing T-cells for an immediate cytotoxic answer [5].

Basically, the synNotch receptor carries a scFv directed against a given antigen, and this scFv can be exchanged without influencing the general architecture of the synNotch receptor. Importantly, synNotch receptors also contain a cytosolic domain that is cleaved upon receptor stimulation. This γ-secretase-dependent cleavage can cause ligand-independent activation [3,6,7] of the SynNotch receptor, but enhanced synthetic notch receptors (esNotch) are reported to significantly reduce the ligand-independent activation [8]. This intracellular domain can be substituted with various proteins that interact with compounds of the cytoplasm or the nucleus [3]. By applying an artificial transcription activator to the synNotch receptor, direct initiation of target gene expression can be achieved. Additionally, the synNotch receptor needs no further costimulatory activation. Therefore, it can be expressed on a broad variety of cell types and is not limited to cells of the lymphatic lineage.

It has been shown that numerous different scFvs and signaling proteins can be used in synNotch receptors and that a combination of the two, used in a multiplexing approach, would allow for the generation of complex signaling cascades [3,6,9]. Furthermore, the use of different synNotch receptor systems on one cell can increase the recognition potential of the cell and thus the specificity of the system [9].

In this way, receptor stimulation can trigger different outcomes, e.g., cell-intrinsic induction of apoptosis or the release of cytotoxic mediators to neighboring cells. In basic research and preclinical studies, the use of fluorophores for fast and easy readout is well-established, and if a long-term stable reporter fluorophore is chosen, information about receptor activation can be temporarily stored and read out at a later time point. As a permanent cellular reporter system, several concepts using genomic DNA as a storage medium have been described [10,11,12], and it seems highly appealing to combine the transient synNotch receptor activation with such a permanent DNA editing-based reporter system for studying differentiation in stem cell research or developmental biology.

Initially, we aimed to investigate cells expressing synNotch receptors with different antigen-binding sites and scFv affinities to evaluate the most suitable conditions for activation of the synNotch receptor. On the basis of our results, we explored whether synNotch receptor-expressing cells could convert activation into permanently stored information through CRISPR/Cas9-mediated gene editing at a specific DNA locus. To this end, a transgenic cell line expressing a synNotch receptor with a cytosolic tetracycline transactivator (tTA, regulator of transcription activation) domain was generated. Additionally, a response element consisting of a Cas9 nuclease carrying a guide RNA (gRNA) under the control of a tTA-inducible promoter was introduced into the cell line. By harvesting the cells after synNotch receptor activation, we were able to detect indel formation at the targeted DNA locus. With these experiments, we provide proof of concept for the conversion of a transient receptor activation signal into permanent information encoded in a cell’s DNA.

## 2. Materials and Methods

### 2.1. SynNotch Receptor and Response Element

The investigated SynNotch receptors were originally designed by the lab of Wendell A. Lim [3] and obtained via Addgene (Watertown, MA, USA). The lentiviral vector constructs pHR_SFFV_LaG17_synNotch_TetRVP64 (Addgene Plasmid #79128), pHR_PGK_antiCD19_synNotch_Gal4VP64 (Addgene Plasmid #79125), pHR_PGK_antiHer24D5-3_synNotch_Gal4VP64 (Addgene Plasmid #85422), pHR_PGK_antiHer24D5-5_synNotch_Gal4VP64 (Addgene Plasmid #85423), and pHR_EGFPligand Addgene Plasmid #79129) were used to generate infectious lentiviral particles using a third-generation self-inactivating lentiviral backbone (pRRL.PPT.SFFV.pre) [13]. This backbone was also modified to generate the response elements, where either five copies of the Gal4 DNA binding domain (DBD) binding motive (GGAGCACTGTCCTCCGAACG) or six copies of the tetracycline response element (TCCCTATCAGTGATAGAGA) were cloned 5′ to a minimal cytomegalovirus (CMV) promoter. For quantification, the dsRed-Express gene was placed under the control of the inducible CMV_min_ promoter. For gene editing experiments, a lentiviral construct expressing the gRNA (scrambled or CXCR4 targeting) under control of the U6 promoter was used. This construct also expressed the Cas9 nuclease, which is linked via a 2A self-cleaving peptides from thosea asigna virus T2A site to Puromycin-*N*-Acetyltransferase (PAC) for selection. The expression was under control of the spleen focus-forming virus (SFFV) promoter (positive control) or the tTA inducible CMV_min_ promoter. For tTA release, the pHR_SFFV_LaG17_synNotch_TetRVP64 (Addgene Plasmid #79128) synNotch receptor was used.

### 2.2. Cell Culture Conditions

If not described differently, HEK293 cells were cultivated in Dulbecco′s Modified Eagle′s Medium (DMEM)-High Glucose Media (Life Technologies; Carlsbad, CA, USA)) supplemented with 10% fetal bovine serum (FBS) (Biochrom; Berlin, Germany), 1% PenStrep (Biochrom, Germany), and 1 × nonessential amino acids (Gibco, Germany) at 37 °C and 5% CO_2_. Cells were passaged by adding 0.5 × Trypsin-ethylenediaminetetraacetic acid (EDTA) (Life Technologies; Carlsbad, CA, USA) after washing with phosphate buffered saline (PBS) (Life Technologies; Carlsbad, CA, USA) centrifuged at 150× *g* and reseeded with a density of 20,000 cells/cm^2^ every 96 h on plastic dishes (TPP Techno Plastic Products AG, Trasadingen, Switzerland). Lymphoblastoid Cell Line (LCL) cells were kindly provided by the lab of R. Stripecke (Regenerative Immune Therapies Applied, Hannover Medical School, Germany). In brief, human B cells were immortalized with the EBV-B95.8/GFP laboratory strain and the best growing cell lines were selected and confirmed for CD19 expression. Cells were cultured in RPMI media (Gibco, Germany) containing 10% FBS. The suspension cells were centrifuged at 200× *g* for 5 min and seeded with a density of 1 × 10e6 cells/mL. The SK-BR3 cell line was cultured in McCoy’s 5A media (Gibco, Germany) containing 20% FBS and 1% PenStrep. Cells were passaged by trypsinization when they reached 90% confluency and centrifugation at 200× *g* for 5 min. Cells were seeded with a density of 30,000 cells/cm^2^.

### 2.3. Lentiviral Preparation

For preparation of lentiviral particles, HEK293T cells were seeded at 70% confluence 1 day prior to transient transfection by calcium phosphate precipitation in a 4-(2-hydroxyethyl)-1-piperazineethanesulfonic acid (HEPES)-containing buffer. Cells were transfected with the lentiviral vector plasmid, pcDNA3.GP.4 × CTE (expressing HIV-1 Gag–Pol polyprotein), and pRSV/Rev of pMD.G (encoding vesicular stomatitis virus glycoprotein) in a ratio 5:5:3:1. The medium was changed after 8 h, and transfected cells were incubated at 37 °C and 5% CO_2_. Vector-containing supernatants were collected after 48 h, filtered (0.45 μm), and used for immediate transduction or stored at −80° C for later experiments.

### 2.4. Generation of HEK293 Receiver Cells

To generate receiver cells, HEK293 cells which express the puromycin-resistant mediating PAC transgene for later selection in the experimental setup were used. In a first step, the tTA or Gal4 binding motif 5′ of the CMV_min_ promoter (Response Element; RE) was transduced into HEK293 cells (German Collection of Microorganisms and Cell Cultures GmbH (DSMZ), Braunschweig, Germany). Cells were seeded with a density of 25,000 cells/cm^2^ in a 6-well culture dish (TPP, Switzerland), and 2 mL of viral supernatant containing 10 µM protamine sulfate (Merck, Darmstadt, Germany) was added to the cells. The medium was changed after 24 h and cultured for another 72 h. These HEK293 cells were passaged and reseeded with a density of 25,000 cells/cm^2^ and transduced a second time with either the LaG17_synNotch_TetRVP64 (tTA RE), the antiCD19_synNotch_Gal4VP64, antiHer24D5-3_synNotch_Gal4VP64, or antiHer24D5-5_synNotch_Gal4VP64 (all Gal4 RE) viral supernatant (2 mL) containing 10 µM Protamine sulfate. This double transduced, heterogeneous cell population was cultured for an additional 72 h and was then activated with their corresponding antigen (see below) and sorted by flow cytometry (see below). Of note, the maintenance culture was frequently analyzed for unspecific or spontaneously dsRed-Express expression.

### 2.5. Generation of Clonal Receiver Cell Lines

Clonal cell lines were obtained by serial dilution of all synNotch receiver constructs (GFP-tTA, CD19-Gal4, HER2.3-Gal4, and HER2.5-Gal4). Cell density was set to 0.3 and 0.6 and one cell per 150 µL and plated in 96 well plates (TPP, Switzerland) with 150 µL per well. When colonies were formed, all wells harboring one single colony were passaged into 12-well plates. The resulting clonally expanded cells were activated and analyzed for dsRed-Express expression. The most reproducible clone was used for further experiments.

### 2.6. Activation of HEK293 Receiver Cells

For activation of LaG17_synNotch HEK293 cells, a GFP sender cell line expressing a membrane-bound GFP (GFP sender cells, see below) was used. For activation of antiCD19_SynNotch HEK293 cells, the LCL cell line and for activation of antiHer24D5-3_SynNotch and antiHer24D5-5_synNotch SK-BR3 cell line was used. Sender and receiver cells were dissociated, and different ratios (see results) of synNotch expressing receiver cells were mixed with their corresponding antigen expressing sender cells in 200 µL DMEM-HG media and incubated in a 1.5 mL tube. HEK293 cells were used as a control for unspecific activation in all experiments. After different time points (see results), all cells were seeded with a density of 40,000 cells/cm^2^ in HEK293 media containing 3 µg/mL puromycin and cultured for further 48 h with daily media change. Cells were trypsinized and analyzed by flow cytometry.

### 2.7. Flow Cytometry

Cells were harvested by trypsinization and centrifuged for 5 min at 200× *g*. The cell pellet was resuspended in Fluorescence Activated Cell Sorting (FACS) buffer (PBS containing 1% FBS and 10 mM EDTA) with a density of 4 × 10e6 cells/mL and filtered (70 µm). For analyses of activation experiments (time and ratio), cells were directly analyzed using the LSRII (Becton, Dickinson and Company (BD) Biosciences; San Jose, CA, USA) and FlowJo v10 Software. For analyses of single clone experiments, cells were directly analyzed using the CytoFlexS (Beckman Coulter; Brea, CA, USA) and CytExpert Software. For sorting (synNotch receptor receiver cells and GFP sender cells), cells were stained with α-myc Tag antibody (Cell Signaling #2233; 9B11; Danvers, MA, USA), washed with PBS, and sorted using FACSAria Fusion (BD Biosciences, San Jose, CA, USA).

### 2.8. GFP Sender Cells

HEK293 cells were seeded (20,000 cells/cm^2^) one day prior and were transduced with lentiviral vector particles encoding a membranous GFP (pHR_EGFPligand Addgene Plasmid #79129) in DMEM-HG containing 10 µM protamine sulfate. The medium was changed after 24 h, and cells were cultured for additional 5 days. Cells were harvested and sorted for GFP-positive cells (see flow cytometry).

### 2.9. SynNotch-Mediated Gene Editing Approaches

For all synNotch mediated gene editing experiments, HEK293 cells carrying the pHR_SFFV_LaG17_synNotch_TetRVP64 (Plasmid #79128; Addgene, Watertown, MA, USA) synNotch were generated as described above. The bulk population was sorted using the α-myc Tag antibody (Cell signaling #2233; 9B11) and expanded. In a next step, these cells were transduced with lentiviral vector particles for the following constructs; positive control (CXCR4 gRNA under control of U6 promoter and an SFFV-driven hspCas9_T2A_PAC), negative control (scrambled gRNA under control of U6 promoter and an SFFV-driven hspCas9_T2A_PAC), and experimental construct (CXCR4 gRNA under control of U6 promoter and a tTA-VP64 inducible CMV_min_-driven hspCas9_T2A_PAC) [13].

### 2.10. Gene Editing Experiments

After lentiviral transduction of HEK293 cells with either the positive or negative control, transduced cells were selected for 72 h with medium containing 2 µg/mL puromycin. For synNotch-mediated gene editing experiments, 7.5 × 10^5^ GFP sender cells and 7.5 × 10^5^ synNotch green fluorescent protein (GFP) expressing and regulator of transcription activation (tTA)- inducible (GFPtTA) receiver cells were incubated together for 1 h in 300 µL media. After incubation, all cells were seeded in a 6-cm dish and cultivated for 24 h. For selection of activated synNotch GFPtTA receiver cells, the medium was changed, containing 2 µg/mL puromycin, and cells were incubated for ab additional 72 h. All selected cells were harvested, and the genomic DNA was isolated (Merck #G1N350; Merck, Darmstadt, Germany) for subsequent sequencing.

For cell cycle modulation, synNotch GFPtTA receiver cells (CXCR4 sgRNA under control of U6 promoter and a tTA-VP64 inducible CMV_min_ driven hspCas9_T2A_PAC) were either cultivated under low FBS condition (1.5%) for three weeks or no FBS for 72 h or were treated with 100 ng/mL nocodazole (Merck #M1404) for 18 h to induce G1-phase arrest (low and no FBS condition) or G2/M-phase arrest nocodazole (Merck #M1404; Merck, Darmstadt, Germany)). Equal amounts (500,000 cells) of treated GFP-tTA receiver cells and GFP sender cells were incubated for 1 h at 37 °C for synNotch activation. After activation, cells were plated and incubated for 24 h at 37 °C and 5% CO_2_ followed by puromycin selection for additional 96 h. Cells were harvested, genomic DNA were isolated (Merck #G1N350), and the CXCR4 gene was amplified (Agilent #600675) and sequenced (Sanger sequencing by Seqlab; Göttingen, Germany). Obtained chromatograms were analyzed using standard settings of TIDE analysis (https://tide.deskgen.com). The gene editing scheme depicted in Figure 9 was created using the biorender.com application.

### 2.11. Detection of Gene Editing Events

For DNA sequencing of the targeted CXCR4 gene locus, 100 ng of template was amplified using 2.5 pmol/µl primer (CXCR4_forw. 5′-CACTTCAGATAACTACACCGAGG-3′ and CXCR4_rev 5′-CTCAGAGGTGAGTGCGTGCTG-3′) and 1 U Polymerase (Agilent #600675; Agilent Technologies, Santa Clara, CA, USA). The amplicon was purified via 1% agarose gel, extracted (Qiagen #28115; Qiagen, Hilden, Germany) and directly sequenced (Sanger sequencing, Seqlab, Göttingen, Germany). Obtained chromatograms were analyzed using standard settings of TIDE analysis (https://tide.deskgen.com).

To further investigate the gene editing efficiency on the amplicon level, purified PCR products were cloned into TOPO^®^ Vector (#450245; Thermo Fisher Scientific, Waltham, MA, USA), plated on Kanamycin containing agar plates (100 µg/mL), and incubated at 37 °C overnight. Colonies were picked and expanded in kanamycin containing (100 µg/mL) Luria-Bertani (LB) media for an additional 24 h. Plasmid DNA was isolated (Qiagen, #27106), checked for correct insert via EcoRI digest (#R3101L; New England Biolabs, Ipswich, MA, USA), and analyzed on a Midori green stained (#MG04; Nippon Genetics, Koraku, Japan) 1% agarose gel. Positive clones were directly sequenced (Sanger sequencing, Seqlab Göttingen).

For an enzymatic detection of synNotch-mediated indel formation, the Surveyor^®^ Mutation Detection Kit (Integrated DNA Technologies, #706020) was used according to the manufacture’s protocol. In brief, the CXCR4 gene was amplified (Agilent #600675) and purified via extraction from a 1.0% agarose gel (Qiagen, #28706), and 400 ng of obtained DNA from gene editing experiments was annealed with the untreated control. Half of the heteroduplex DNA was digested with surveyor nuclease for 1 h. Internal controls were equally processed. Digested and undigested DNA were analyzed on a 1.5% agarose gel stained with Midori Green (Nippongenetics, #MG04). A 1 kBp ladder (Thermo Scientific; # SM0313) was used as a size standard.

### 2.12. Statistical Analyses

Statistical analyses were performed using a one- or two way analysis of variance (ANOVA). Tukey’s method was employed as a post hoc test. The results shown were obtained by at least three measurements (mean ± standard deviation (SD)) from independent biological experiments. Differences were considered statistically significant at *p*-values below 0.05 and are marked with asterisks (* *p* < 0.05, ** *p* < 0.01, *** *p* < 0.001, and **** *p* < 0.0001).

## 3. Results

### 3.1. Generation of synNotch Receiver Cell Lines

As different scFvs show different binding affinities [14,15], we first tested synNotch receptors raised against different antigens. For our initial experiments, an artificial scFv domain recognizing a membrane-bound GFP was chosen.

Two further antigens that are broadly used in CAR therapy, CD19 (used against hematopoietic malignancies) and HER2 (used against neuroblastoma), were chosen for further experiments. For HER2, two distinct scFvs that bind distinct epitopes of the antigen were investigated. Additionally, different transactivators coupled to the synNotch receptor domains tTA and Gal4 were used to induce downstream gene expression. The corresponding binding sequence was cloned in front of the cytomegalovirus minimal promoter (CMV_min_) promoter (Figure 1a) and was verified by Sanger sequencing. Two distinct HEK293 cell lines carrying either the tTA response element or the Gal4 response element were generated. The following synNotch receptor constructs were used in the experimental setup: GFP-tTA, CD19-Gal4, HER2.3-Gal4 (low affinity), and HER2.5-Gal4 (high affinity). The bulk populations generated for each synNotch receptor were activated by cell lines expressing the corresponding antigens. Activated dsRed-Express-expressing cells were sorted for dsRed-Express expression to obtain an inducible population for each type of synNotch receptor (Figure 1b and Table 1). A second sort was performed to confirm synNotch receptor expression and to rule out false positive events caused by spontaneous activation or gene transcription of pre-integration complexes (Figure 1c,d and Table 2). The sorted cell populations were expanded and used as sources of synNotch receiver cells for subsequent experiments.

### 3.2. Evaluation of Most Suitable Conditions for SynNotch Receptor Activation

In previous reports, the synNotch-expressing receiver cells were permanently co-cultured with antigen-presenting sender cells [3,9]. However, we wanted to determine conditions for proper synNotch activation and downstream gene expression by the most suitable amount of sender cells and the ideal co-culture time. To this end, we initially used different ratios of sender (antigen-expressing) cells and receiver (synNotch receptor-expressing) cells. We could demonstrate that a ratio of one sender cell to five receiver cells already leads to a clearly quantifiable expression of the synNotch-mediated gene expression (Figure 2). However, increasing the proportion of sender cells increased the dsRed-Express expression (Figure 2). In case of the GFP-tTA synNotch construct, the amount of positive cells increased from 7.6% ± 1.6% (sender–receiver ratio 5:1) to 36.7% ± 1.8% (for 1:1 ratio; 4.82-fold). A further increase of the ratio resulted only in a minor improvement from 36.7% ± 1.8% (1:1 ratio) to 40.1% ± 1.1% (5:1 ratio; 1.09-fold) (Figure 2 and Table 2). Similar results were obtained for the CD19-Gal4 and HER2.5 constructs (Figure 2 and Table 2). Unspecific activation of the different synNotch receiver cell lines upon co-culture with HEK293 cells under the same conditions did not result in detectable dsRed-Express expression (Appendix A). Thus, for all subsequent experiments, a ratio of one sender cell to one receiver cell was used to achieve robust synNotch receptor activation and downstream gene expression.

Next, we evaluated the co-cultivation time period required to achieve synNotch receptor-mediated reporter fluorophore expression. Sender and receiver cells were incubated together for the indicated times before the cells were re-plated at low density to suspend cell–cell contact. Then, the cells were cultivated in selection media containing puromycin to remove the sender cells. As above, all experiments were performed with a ratio of one sender cell to one receiver cell.

We observed minor synNotch activation after only 10 min (Figure 3). The amount of dsRed-Express-expressing receiver cells increased steadily with increasing incubation time. With the GFP-tTA construct, the amount of dsRed-Express-positive cells increased from 7.3% ± 1.3% after 10 min to 24.8% ± 1.5% after 30 min (3.49-fold increase). Incubation for 60 min increased the dsRed-Express-positive cell population to 31.3% ± 1.7%, and incubation for 120 min resulted in 37.3% ± 1.5% dsRed-Express-positive cells. However, extended co-cultivation (180 min) did not result in a further increase in dsRed-Express-positive cells (36.3% ± 2.6%), as depicted in Figure 3 (upper row). Similar results were obtained for the CD19-Gal4, HER2.3-Gal4, and HER2.5-Gal4 constructs (Figure 3, 2nd, 3rd, and last rows); in all cases, a substantial amount of dsRed-Express-positive cells were detectable after 60 min of activation. Importantly, co-culture of the four investigated cell lines with HEK293 cells bearing no synNotch construct did not result in demonstrable reporter activation, as measured based on dsRed-Express fluorescence (Appendix A). On the basis of these experiments, we determined that an activation time of 60 min and a ratio of one sender cell to one receiver cell were most suitable for achieving robust synNotch receptor-mediated activation of the reporter fluorophore for subsequent flow cytometry quantification (Table 2).

After clonal expansion of targeted cells within the receiver cell population, we were able to detect enhanced activation of synNotch-mediated fluorophore expression (Figure 4). We observed a significant increase from 34.8% to 73.6% (*p* = 0.0021) for the GFP-tTA construct, from 18.6% to 56.1% (*p* = 0.0067) for CD19-Gal4, from 17.4% to 53.1% (*p* = 0.0043) for HER2.3-Gal4, and from 24.3% to 43.9% for HER2.5-Gal4 (Table 2).

### 3.3. SynNotch-Mediated Gene Editing

To investigate the feasibility of synNotch receptor-mediated gene editing, we chose the GFP-tTA synNotch receptor, which cleaves the tTA transcriptional activator upon activation. For the reporting element, we designed a construct that harbors six tTA-binding motifs in front of a minimal CMV promoter-driven Cas9 transgene and a U6 promoter-driven CXCR4 gRNA (Figure 5a). A spleen focus-forming virus (SFFV) promoter-driven Cas9 and a U6 promoter-driven scramble gRNA served as a negative control (Figure 5b), whereas an SFFV promoter-driven Cas9 combined with a U6 promoter-driven CXCR4 gRNA served as a positive control (Figure 5c).

HEK293 receiver cells carrying one of these constructs were co-cultured with the membrane-bound GFP-expressing cells as described above (ratio 1:1, 60 min) and further propagated for 96 h before genomic DNA was harvested for subsequent analyses of Cas9-mediated genome editing events. Primers designed within the CXCR4 locus were suitable for all analyses performed (Figure 5d).

For a more qualitative approach, we analyzed synNotch receptor-mediated gene editing using a surveyor nuclease assay. By hybridizing genetically unmodified CXCR4-amplified DNA with DNA from putatively targeted cells, duplex DNA was generated. Based on mismatches generated by gene editing events, a specific double-strand cut should occur near the protospacer adjacent motif(PAM) sequence. To ensure specific nuclease digestion, a reference homoduplex (CTRL1, Figure 6) that did not have a mismatch was also treated with the nuclease (CTRL1 digest, Figure 6). A reference heteroduplex harboring a single mismatch (CTRL2, Figure 6) served as a positive control. After nuclease treatment of the positive control (CTRL2, Figure 6), the expected bands of 217 and 416 bp appeared (CTRL2 digest, Figure 6), indicating that the nuclease conditions were appropriate. The CXCR4 gene was amplified from genetically unmodified HEK293 cells. The inactivated control cells used in our experiments (HEK and HEK digest, Figure 6) did not show any digestion bands, indicating a non-mismatched homoduplex. Similarly, the activated but scrambled RNA-guided control cells (SCR, Figure 6) did not show an extra band after nuclease treatment (SCR digest, Figure 6), indicating that the scramble gRNA was not able to recruit the Cas9 nuclease to the CXCR4 PAM sequence. When the SFFV promoter-driven Cas9 was expressed constitutively (SFFV, Figure 6), the expected digestion bands appeared around 350 and 585 bp after nuclease treatment (SFFV digest, Figure 6), indicating that Cas9 was recruited to the CXCR4 PAM sequence by the specific gRNA and that indel formation occurred to a certain extent. Finally, in our synNotch receptor-mediated gene editing experiments (experiments 1–3, Figure 6), all nuclease-treated heteroduplex DNAs presented the same digestion pattern (experiments 1–3 digest, Figure 6), indicating that, after synNotch activation, the tTA element was able to bind to the tTA response element and to initiate Cas9 expression.

Next, we investigated the frequency of gene editing events using the tracking of indels by decomposition (TIDE) web tool for quantitative analysis. First, we analyzed indel formation frequencies within the CXCR4 amplicons caused by spontaneous mutations within the HEK293 cell population, representing background effects influencing the sequencing results. To do this, we amplified and sequenced genomic HEK293 DNA from cells of different passages. The variation between the samples gave rise to an overall indel efficiency of 0.9%, although *p* < 0.001 was not achieved (Table 3). Additionally, U6 PolII promoter-driven scrambled gRNAs (Figure 5a) in combination with SFFV-driven constitutive Cas9 expression showed only minor unspecific indel formation (2.4%, Figure 7a and Table 3). The positive control (Figure 5b, CXCR4-targeting gRNA and SFFV-driven Cas9 expression) showed an indel formation frequency of 47.6% in the investigated bulk population (Table 3). In three independent experimental settings, synNotch-triggered Cas9-mediated gene editing resulted in indel formation frequencies of 34.5%, 41.3%, and 45.4% (Figure 7b and Table 3). To verify that the observed events were not attributable to leaky CMV activation as reported previously [12], we analyzed a further nonactivated control expressing the CXCR4 gRNA and a tTA-inducible CMV_min_, which showed a low indel formation frequency of 2.7% (Table 3). Additionally, unspecific activation of the synNotch receptor by HEK293 cells bearing no antigens (HEK^empty^) resulted in a low indel formation frequency of 3.7% (Table 3).

As a complementary approach to the TIDE analysis in experiment 1, we checked single amplicons of the amplified CXCR4 gene by transfer of PCR products into a cloning vector and subsequent Sanger sequencing of a subset of transfected clones. Seven out of 12 analyzed sequences exhibited indel formation (Table 4), which correlates well with the result of the TIDE analysis (47.9%). With these data, we were able to confirm the insertion of one base in four clones and a deletion of three and five bases in one and two clones, respectively, which is quite close to the TIDE analysis prediction (Figure 8).

Most established cell lines, such as the HEK293 cells used in our study, are rapidly dividing cells which facilitate gene editing approaches in comparison with less actively dividing cells in living organisms. Homolog-directed repair requires an active cell cycle and thus dividing cells, whereas non-homolog end joining-mediated repair is possible even in nondividing, postmitotic cells [16,17]. Therefore, we analyzed the frequency of synNotch-triggered non-homolog end joining-mediated gene editing with and without various interfering modifications to the cell cycle. First, we cultivated receiver cells under low-FBS conditions (1.5% FBS) or without FBS to induce G1 phase arrest. To induce G2/M phase arrest, cells were treated with nocodazole. By performing synNotch-triggered Cas9-mediated gene editing experiments as described above, we were able to detect indel formation in all experiments by TIDE analysis (Table 5). The frequency of indel formation in cells arrested in G1 phase ranged from 31.4% (no FBS) to 37.4% (low FBS), whereas the nocodazole-treated cells exhibited an indel formation frequency of 34.4%. Interestingly, the efficiency of HEK^empty^-stimulated control receiver cells (3.7%, 3.3%, and 3.1%) was comparable with that of HEK-stimulated control receiver cells in which the cell cycle was not arrested (3.7%; Table 3).

## 4. Discussion

By rational assembly of diverse receptor elements, intracellular signaling cascades, and gene transcription regulators, synthetic biology can be used to design artificial transduction pathways. The combination of various elements originating from different species minimizes interference with physiological host cell signaling. Modular receptors designed in this way can sense a broad spectrum of natural and synthetic extracellular signals that could be converted to user-defined cellular responses for research and, eventually, therapeutic applications [3,6,9,18,19,20]. In essence, a physiological, disease-related, or synthetic epitope can activate a receptor that converts this signal into a response (Figure 9). Such a response could manifest as the expression of a reporter signal (e.g., a fluorophore such as GFP or dsRed-Express), as a forwarding signal (e.g., a transcriptional regulator such as tTA or Gal4), or as a direct cellular response (e.g., a cytotoxic event or differentiation) [3]. So far, synthetic receptors have been intensively studied in preclinical and clinical applications, where CARs were linked to immune cells, such as T cells or NK cells, which mediate a direct cytotoxic response [21]. However, these settings are not designed to capture information about the amount of antigens present (e.g., to quantify the progress of tumor formation) or about the localization and context of the antigen contact (e.g., to discriminate primary neoplasia from metastases or to observe tumor-inherent mutations). These challenges might be approached with much more sophisticated cellular systems in which different synNotch receptors can be used in a multiplex setup to recognize several antigens and to report this information instead of providing a cytotoxic answer.

By exploiting selected scFvs for antigen recognition and robust transcriptional activator systems, we were able to demonstrate the time- and dose-dependency of the synNotch receptor activation-mediated fluorophore expression of receiver cells. In our experiments, four different synNotch receptor-expressing cell lines were co-cultivated with corresponding antigen-expressing cells in close proximity under than in rather static conditions. We observed that a low ratio of antigen-expressing sender cells (20% of the cell population) was sufficient to activate the synNotch receptor-expressing receiver cells in relevant numbers and that higher ratios led to higher numbers of activated cells. However, any further increase in activated dsRed-Express fluorescent cells was limited when increased ratios of sender cells were applied, and we conclude that this might have been due to the fact that not all receiver cells were seeded close enough to an antigen-expressing sender cell. Similar considerations are applicable to activation time. Although a relatively short incubation time of about 10 min was enough to detect receptor activation, increasing the incubation time to 60 min resulted in a higher frequency of activated cells. However, stimulation for 120 and 180 min did not result in a consistent further increase in the frequency of fluorophore-expressing cells. Interestingly, the fluorescence intensity also did not increase further in these experiments, suggesting a saturation stage in our experimental setup.

This work aimed to convert the transient synNotch receptor activation to an event of much longer duration than temporary reporter fluorophore expression. We considered the induction of genetic marks by CRISPR/Cas9 technology as the most versatile approach to storing molecular information in a cell’s DNA for later readout (Figure 9). This assessment is supported by a previous study [18] in which an inactive dCas9-derivative fused to a VP64 transactivator was used to specifically induce reporter gene expression (mCherry) after receptor activation.

Even if synNotch receptors are reported to have a given amount of ligand-independent activation (LIA) [3,6,7,9], one could assume that this drawback may be reduced by different means. As reported by Yang et al., an additional hydrophobic sequence (named as RAM7) which is present in the native notch receptor significantly reduces the LIA [8]. Such an enhanced synthetic notch receptor (esNotch) did not exhibit altered antigen-induced activation or surface expression. Another study could demonstrate that an apelin-based synNotch receptor (AsNRs) was able to specifically detect neovascular endothelium in adult tissues, despite a small amount of LIA [4]. Similarly, we report in our study only a minor rate of LIA (0.46%–1.13%) as presented in Appendix A) and we suggest that further improvements of synNotch receptor design may eventually result in a robust detection systems for future applications in biomedical research.

Another recent study using the Cas9:p300 fusion protein showed that the expression of a given gene could be either activated or suppressed after synNotch receptor activation [7]. Indeed, the authors of this study could demonstrate synNotch-mediated indel formation but observed a high unspecific background activity or LIA when the Cas9:p300 expression was triggered via a Gal4-UAS-system. Hence, the combination of the synNotch receptor with a Tet-on system that was controlled by both CD19^+^ cells and Doxycycline (Dox) treatment was necessary to achieve efficient indel formation under conditions that exhibited only minor background activity. In our approach, we took advantage of a synNotch receptor-triggered tTA (tetracycline transactivator protein) in a Tet-Off background. In this approach, the expression of the Cas9 is independent from Dox supplementation because tTA-mediated transcription is activated in the absence of Dox. Interestingly, in all of our experimental setups, we observed only minor unspecific expression of the dsRed-Express fluorophore (Appendix A) or background indel formation (Table 3) in the absence of the respective receptor stimulus. Furthermore, unspecific stimulation in cells not expressing the appropriate antigen led to negligible dsRed-Express expression (Appendix A) and indel formation (Table 3 and Table 5). This indicates that the applied synNotch effector system resulted in robust and rather specific signaling events under our experimental conditions. However, upon activation of the synNotch receptor, we observed robust gene editing events, i.e., indel formation, and our data clearly suggest that these gene editing events occurred at similar frequencies (34.5%, 41.3%, and 45.4%) to the expression of the reporter dsRed-Express in our initial set of experiments (36.7% ± 1.8%). However, the indel formation only recapitulates the synNotch receptor-triggered Cas9-mediated DNA cleavage after contact with the respective antigen and could not record more detailed information about the duration of the antigen contact or other properties of the targeted cell or structure. More sophisticated approaches, as reported by Farzadfard et al. [12], could be used to specifically record such information. If different signals are processed via different synNotch receptors, the logical sequence of those signals can be captured. A DNA-based Ordered Memory and Iteration Network Operator (called DOMINO) operator technology allows one to determine whether different signals were detected independently (“AND” gate) or in a given chronology (“AND THEN” gate). Furthermore, if there is repetition of the signals over time, this repetitive activation can also be stored in a logical manner (“AND AFTER TIME THEN” gate). By combining this technology with the synNotch receptor effector system in future research, a broad variety of intercellular communication might be monitored.

Besides studies using scFvs raised against reporter proteins or oncogenes as used in the present study, one could envision a reporter cell system that allows a deeper understanding of cellular interactions in more complex cell aggregates or in developing embryos. For instance, during the generation of organoids from pluripotent stem cells, a variety of intrinsic and extrinsic signals steer the formation of complex multicellular structures [22,23,24]. In this scenario, the maturation of cells within a complex structure could be monitored if a membrane-bound epitope is expressed at a given developmental stage. Thus, adapting synNotch receptors to recognize critical cellular interactions during key developmental stages could extend our understanding of in vitro differentiation processes. It may be possible to apply similar systems in genetically modified animals to record key cellular interactions during organ formation or other developmental/regenerative processes. Additionally, a synNotch receptor system could be used in a diagnostic manner to report and to putatively quantify degenerative events or early malignant tumor formation at the cellular level.

In conclusion, we provide here proof of concept for the combination of synNotch receptors with CRISPR/Cas-based genetic editing tools to capture information about transient antigen-binding events in various cell culture conditions and to permanently store that information (Figure 9) in the genomic DNA of the receiver cell.

## Figures and Tables

**Figure 1 cells-09-01929-f001:**
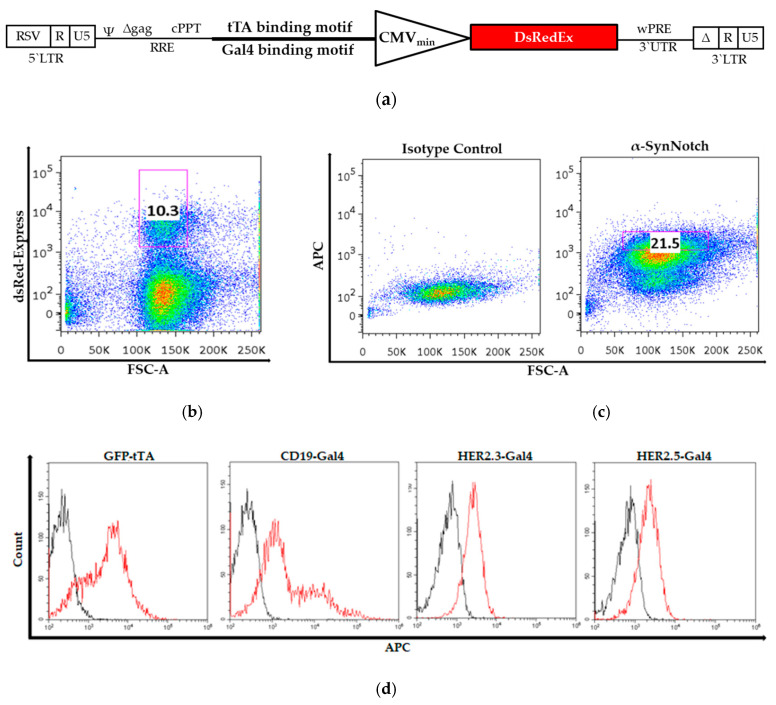
Generation of synNotch receiver cells: (**a**) For reporting gene transcription after synNotch receptor activation, two cell lines were generated by lentiviral transfer of genes carrying transactivator (regulator of transcription activation (tTA) or Gal4) response element. (**b**) After lentiviral transduction, synNotch receptor-expressing cells were activated with their corresponding antigens, and dsRed-Express-positive cells were selected. (**c**) After dsRed-Express expression decreased, a second flow cytometry sorting analysis based on synNotch expression was performed to confirm synNotch expression and to rule out unspecific response element activation. (**d**) The resulting cell lines were analyzed after expansion and prior to further experimentation for synNotch receptor expression. Black line = control, red line = synNotch receptor (α-myc tag).

**Figure 2 cells-09-01929-f002:**
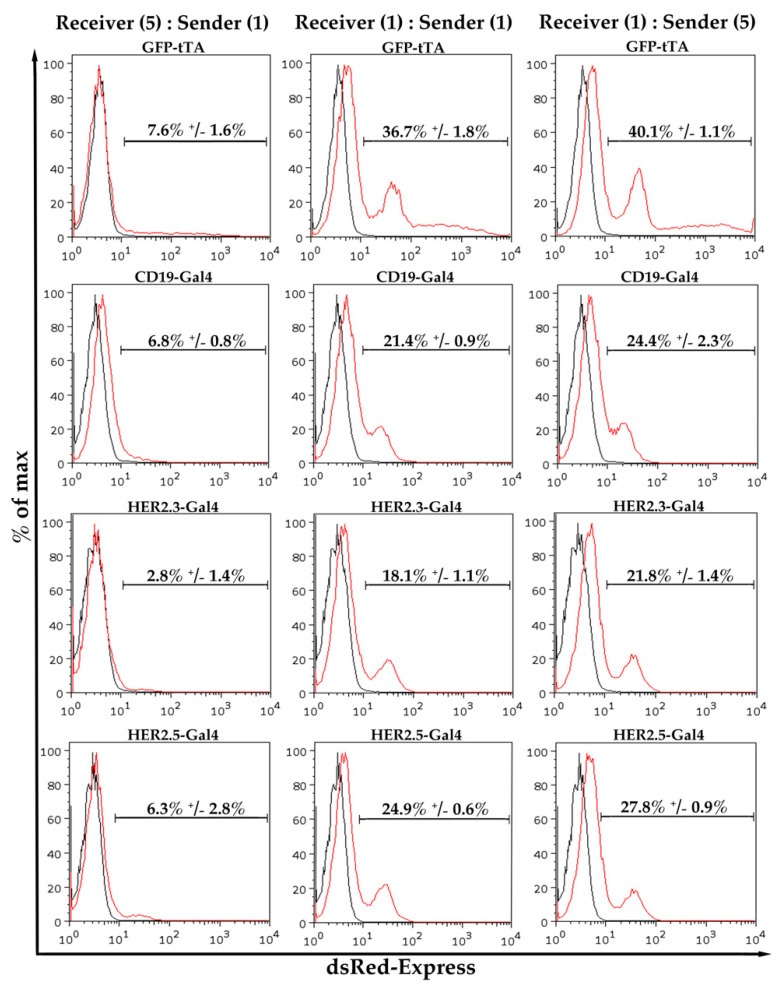
Effect of the ratio of receptor-presenting receiver cells to antigen-presenting sender cells on synNotch receptor activation: To determine the ideal ratio, three different ratios (5:1, 1:1, and 1:5) of synNotch-receptor (targeted against GFP, CD19, HER2.3, or HER2.5) expressing receiver cells were co-culture with their corresponding antigen expressing cells to induce dsRed-Express expression. Receiver cells were quantified for dsRed-Express expression after 48 h by flow cytometry (*n* = 3).

**Figure 3 cells-09-01929-f003:**
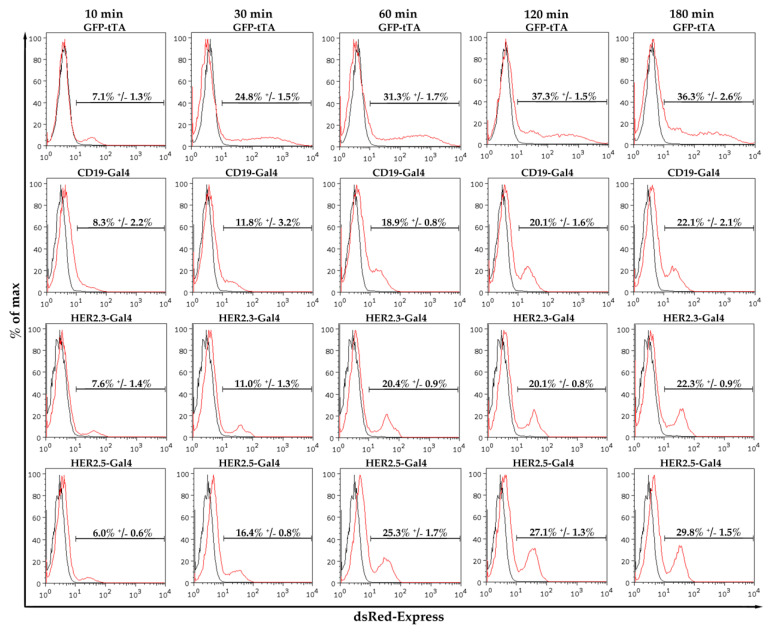
Effect of activation time on synNotch receptor activation: To determine sufficient activation time, synNotch receiver cells were incubated with corresponding antigen-expressing sender cells. After the indicated time points (10, 30, 60, 120, and 180 min), activation was halted. Cells were analyzed 48 h later for dsRed-Express fluorescence (*n* = 3).

**Figure 4 cells-09-01929-f004:**
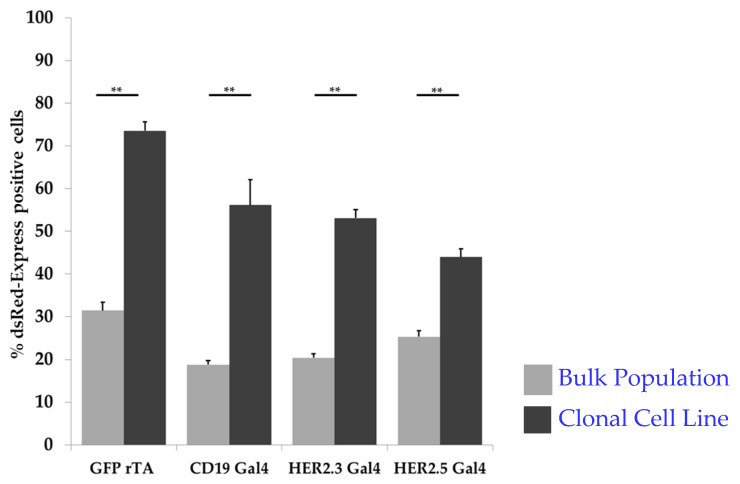
The generation of clonal cell lines further improved the synNotch receptor-mediated fluorophore expression of all four investigated synNotch receptor constructs under the same conditions (sender cell–receiver cell ratio 1:1, 60 min) (*n* = 5,** *p* < 0.01).

**Figure 5 cells-09-01929-f005:**
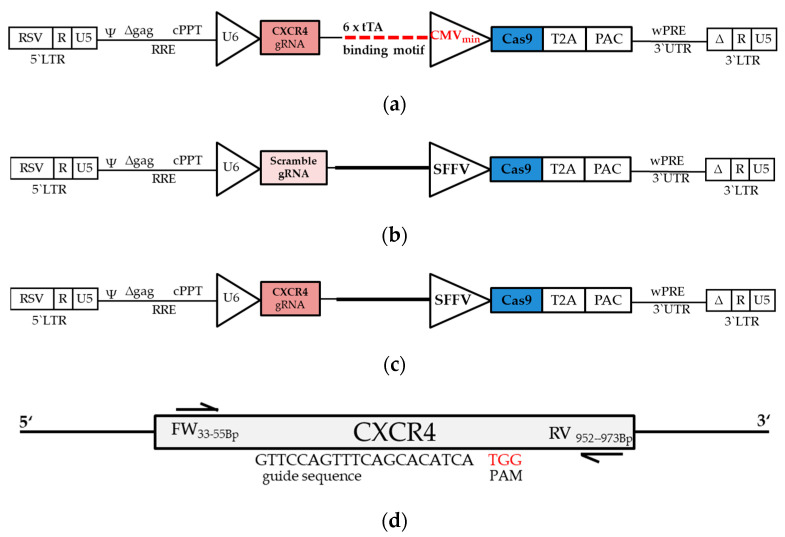
Schematic overview of lentiviral constructs: A scramble (**a**) or CXCR4-targeting (**b**) guide RNA was expressed under the control of the U6 promoter and served as a negative or positive control, respectively. Constitutive (**a**,**b**) or inducible (**c**) expression of hspCas9 was linked via a 2A self-cleaving peptides from thosea asigna virus (T2A) site to puromycin-*n*-acetyltransferase for later selection. (**d**) The amplicon length of the CXCR4 gene was sufficient for all analyses after gene editing.

**Figure 6 cells-09-01929-f006:**
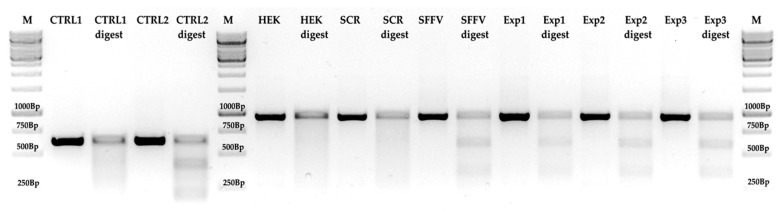
Surveyor nuclease assay of synNotch-mediated gene editing: Control homoduplex DNAs (CRTL1) did not give rise to additional bands after digestion (CTRL1 digest). Control heteroduplex DNA (CTRL2) showed the expected bands. Homoduplex HEK CXCR4 DNA (HEK) and heteroduplex CXCR4 DNA from receiver cells harboring scrambled guide RNA (SCR) did not show extra bands after nuclease treatment (HEK digest, SCR digest). Heteroduplex CXCR4 DNA from receiver cells harboring either the constitutively expressed Cas9 (spleen focus-forming virus (SFFV)) or the tTA-inducible CMV_min_ (experiments 1–3) showed the same band pattern after nuclease treatment (SFFV digest; experiments 1–3 digest). A 1 kbp ladder was used for size control on 1.5% analytic agarose gel.

**Figure 7 cells-09-01929-f007:**
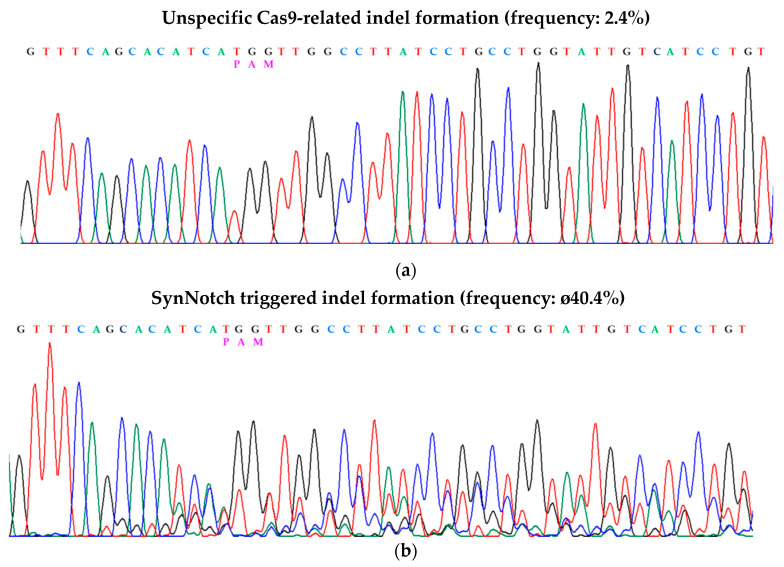
SynNotch receptor-triggered CRISPR/Cas9-mediated gene editing: Partial chromatogram and TIDE-based indel quantification of synNotch receptor–triggered CRISPR/Cas9-mediated gene editing in inactivated HEK293 receiver cells (**a**) and synNotch receptor-triggered CRISPR/Cas9-mediated gene editing in activated HEK293 receiver cells (**b**).

**Figure 8 cells-09-01929-f008:**
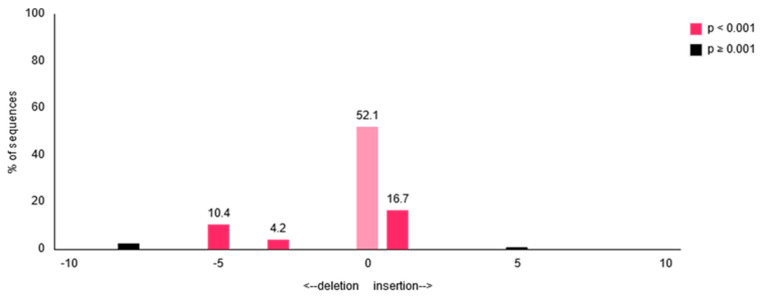
Graphical overview of synNotch-mediated indel formation within the CXCR4 gene based on chromatogram files analyzed using TIDE.

**Figure 9 cells-09-01929-f009:**
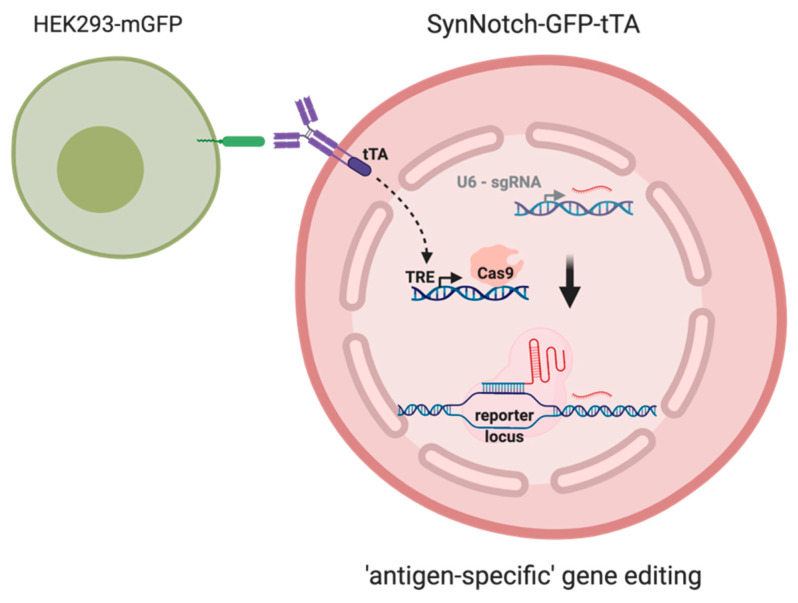
Graphical representation of the presented work: Upon synNotch receptor stimulation, the cytosolic domain of the receptor (tTA, tetracyclin transactivator) gets released and is able to activate the expression of the Cas9 via the Tet-responsive elements (TREs) in the promoter sequence. Guided by constitutively expressed gRNA, the Cas9 initiates a double-strand break next to the protospacer adjacent motif (PAM) sequence. During non-homologous end joining (NHEJ)-mediated DNA repair, a indel formation occurs and can be detected with DNA sequencing approaches. Thus, a transient cellular event could be stored as permanent information in the cell’s DNA.

**Table 1 cells-09-01929-t001:** Results of flow cytometry analyses of synNotch-positive receiver cells.

Construct	% dsRed-Express-Positive Cells	% α-myc-APC-Positive Cells
**GFP-tTA**	10.3	80.2
**CD19-Gal4**	8.4	76.5
**HER2.3-Gal4**	7.3	57.3
**HER2.5-Gal4**	6.8	45.6

**Table 2 cells-09-01929-t002:** Quantification of activation of synNotch receptor systems in bulk populations and clonal cell lines after 60 min of antigen contact and a ratio of one sender cell to one receiver cell.

Construct	% Activation in Bulk Population	% Activation in Clonal Cell Lines
**GFP-tTA**	31.3 ± (−1.7)	73.5 ± 2.1
**CD19-Gal4**	18.9 ± 0.8	56.1 ± 5.9
**HER2.3-Gal4**	20.4 ± 0.9	53.1 ± 2.0
**HER2.5-Gal4**	25.3 ± 1.7	47.3 ± 6.5

**Table 3 cells-09-01929-t003:** Efficiency of synNotch receptor–triggered CRISPR/Cas9-mediated gene editing in three independent experiments (bold) and five control conditions (regular font).

Experiment	Overall Efficiency	Insertion	Deletion
**Experiment 1**	**34.5% (*p* < 0.001)<**	**17.5%**	**17.0%**
**Experiment 2**	**41.3% (*p* < 0.001)<**	**25.9%**	**15.4%**
**Experiment 3**	**45.4% (*p* < 0.001)<**	**30.3%**	**15.1%**
HEK293 cells	0.9% (*p* ≥ 0.01)	0.2%	0.5%
Scramble control	2.4% (*p* ≥ 0.01)	0.1%	2.3%
Positive control	47.6% (*p* < 0.001)<	30.9%	16.7%
Not activated receiver cells	2.7% (*p* < 0.001)	1.3%	1.4%
HEK^empty^ activated receiver cells	3.7% (*p* ≥ 0.001)	1.6%	2.1%

**Table 4 cells-09-01929-t004:** Sequences of single PCR amplicons of synNotch-mediated gene editing via indel formation within the CXCR4 gene.

	PAM	indel
**CXCR4**	GTGTTCCAGTTTCAGCAC-ATCA**TGG**TTGGCCTTATCCTGCCTGGTATT	
**Clone 2**	GTGTTCCAGTTTCAGCAC-ATCA**TGG**TTGGCCTTATCCTGCCTGGTATT	
**Clone 5**	GTGTTCCAGTTTCAGCAC-ATCA**TGG**TTGGCCTTATCCTGCCTGGTATT	
**Clone 6**	GTGTTCCAGTTTCAGCAC-ATCA**TGG**TTGGCCTTATCCTGCCTGGTATT	
**Clone 9**	GTGTTCCAGTTTCAGCAC-ATCA**TGG**TTGGCCTTATCCTGCCTGGTATT	
**Clone 11**	GTGTTCCAGTTTCAGCAC-ATCA**TGG**TTGGCCTTATCCTGCCTGGTATT	
**Clone 12**	GTGTTCCAGTTTCAGCACAATCA**TGG**TTGGCCTTATCCTGCCTGGTATT	**+1**
**Clone 3**	GTGTTCCAGTTTCAGCACAATCA**TGG**TTGGCCTTATCCTGCCTGGTATT	**+1**
**Clone 8**	GTGTTCCAGTTTCAGCACAATCA**TGG**TTGGCCTTATCCTGCCTGGTATT	**+1**
**Clone 7**	GTGTTCCAGTTTCAGCACAATCA**TGG**TTGGCCTTATCCTGCCTGGTATT	**+1**
**Clone 4**	GTGTTCCAGTTTCAGCAC----A**TGG**TTGGCCTTATCCTGCCTGGTATT	**−3**
**Clone 10**	GTGTTCCAGTTTCAGC------A**TGG**TTGGCCTTATCCTGCCTGGTATT	**−5**
**Clone 1**	GTGTTCCAGTTTCAGC------A**TGG**TTGGCCTTATCCTGCCTGGTATT	**−5**

**Table 5 cells-09-01929-t005:** TIDE analyses of synNotch-mediated gene editing in receiver cells in which the cell cycle was arrested at different points.

Experiment	Indel Formation Frequency	Insertion	Deletion
Low FBS	37.4% (*p* < 0.001)	23.3%	14.1%
No FBS	31.4% (*p* < 0.001)	19.0%	12.4%
Nocodazole	34.4% (*p* < 0.001)	26.5%	7.9%
Low FBS; HEK^empty^ stimulated	3.7% (*p* < 0.001)	1.6%	2.1%
No FBS; HEK^empty^ stimulated	3.3% (*p* < 0.001)	1.5%	1.8%
Nocodazole; HEK^empty^ stimulated	3.1% (*p* < 0.001)	2.2%	0.9%

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
