# Peer review of "Synthetic Notch-Receptor-Mediated Transmission of a Transient Signal into Permanent Information via CRISPR/Cas9-Based Genome Editing"

_cells, 2020, doi:10.3390/cells9091929_

Round 1

Reviewer 1 Report

The revised version of the paper presents a significant improvement over the original version. The authors have adequately addressed my concerns. I believe the logic of the paper has improved, the data is better presented, and the figures are constructed well.

While the author do a better job at comparing their results to existing literature, I would still encourage a more full survey of the relevant literature.

Reviewer 2 Report

The authors have addressed all my queries. I have no further questions.

Reviewer 3 Report

The authors still have not made clear of the biological importance of the concept of transforming a transient signal to a permanent signal. If such topic does become important in biological applications, the synNotch activation system and CRISPR-based system seem not to be suitable for such purpose, since both systems have serious fidelity problem. For example, synNotch receptor activation has non-specific leakiness signaling and chances of unresponsiveness, and CRISPR-editing also has a dramatic chance of editing failure. Thus, this system can neither efficiently store transient signal, nor can guarantee that the stored signal is a true reflection of the specific signal. The authors should at least bring this into discussion.

The introduction still have dramatic amount of statements not from this study but are lacking citations. Also, it has redundant background information unrelated with the story, for example, TCR, CAR, etc.

The authors failed to present the data in a precise and efficient way, and this applies to most of the figures. This significantly decreases the readability of the whole manuscript. A lot of the data in the current figures can go to supporting materials. 

Round 2

Reviewer 3 Report

I suggest to publish with the current format.

This manuscript is a resubmission of an earlier submission. The following is a list of the peer review reports and author responses from that submission.

Round 1

Reviewer 1 Report

In the manuscript, the authors present a synthetic tool that enables cells to record transient events in a permanent form that can be read at any later time point. To achieve this they coupled a synthetic variant of the Notch receptor, SynNotch, to sense the signal, with a downstream CRISPR/CAS9 based gene editing system, to permanently store the information. Such a tool has significant implications for studying biological processes as it opens up a path to read the entire history of a cell in addition to its current state.

While the idea of event recording is very important, we believe that this specific proof of principle experiment does not present novel results. Synthetic systems that record events based on signal activation and the CRISPR system have been shown before more generally (e.g. Kipniss et al, Nature Communications, 2017, Frieda et al, Nature  2017) and specifically with the SynNotch system (Huang et al, Protein & Cell, 2020). In addition, the optimization experiments shown in the paper, does not yield a specific new conclusion that might provide new insight. 

Some additional points are as follows:

  1. In section 3.1 the authors construct a fluorescent based system, which they optimize using FACS. It is not clear how this is relevant to the non-fluorescent, Cas9 cell lines. It is further not clear which cell line out of the four generated, the authors use to optimize their protocol in section 3.2. Furthermore, the authors do not explain why they use the conditions optimized for this single cell line across all 4 cell lines, and even for the differently generated Cas9 cell lines. 
  2. The authors optimize their conditions based on the level of fluorescence, however later use the percentage of cells as their quantification. Importantly, the optimization results show that the percentage of positive cells keeps increasing rather than reaching saturation. A more quantitative metric might help to better define this point.
  3. Many plots are not labeled properly, without a legend, and no indication of what is the control condition.

Reviewer 2 Report

Comments on manuscript entitled “SynNotch-Receptor Mediated Transmission of a Transient Signal into Permanent Information via CRISPR/Cas9-Based Genome Editing”

General comment:

In this manuscript, Sgodda et al. investigated the utility of synthetic reporter cells that are able to store transient events with long term memory. By synthesizing an antigen receptor together with a cleavable intracellular domain able to transcriptionally activate a Cas9 nuclease with a specific sgRNA sequence, the authors hypothesized that transient exposure to an activating antigen would lead to genomic aberrations that are stored in memory. The designing of these chimeric antigen receptors, termed SynNotch, are described in this manuscript, alongside the proof-of-concept of its utility. The authors first generated cell lines expressing SynNotch receptors with various antigen recognition domains and cytosolic transcription activation domains. The concentration and duration of transient antigen exposure to elicit a test transcriptional response was then determined. Finally, the authors picked the SynNotch receptor cell line (GFP-rTA HEK293) with the highest transcriptional response to show that their Cas9-sgRNA construct could be transiently transcriptionally activated to leave a significant level of permanent indels when compared to controls.

Major comments

  1. While the premise of the technology is valid and interesting (Using DNA indels as a long-term marker for transient contact), there are definitely downsides to using that in a cellular system, and this is not clearly mentioned in text. The authors should discuss how this technology can applied in cell biology/modelling systems.
  2. As HDR is known to be highly active only in dividing cells (such as HEK293), Cas9 indel generation should theoretically only be as effective as described in the manuscript when used on actively dividing lines such as HEK293. To demonstrate relevance in other biological systems, the authors should compare the HEK293 system to a less actively dividing stem cell line, and show that the system still works as intended.

Minor comments

  1. The language needs to be improved. Currently, the manuscript is very dense in its paragraphing, and there are many run-on sentences that need to either be separated with commas (lines 51, 77 … etc) or made into two sentences entirely.
  2. To improve clarity of the manuscript, the authors may consider to include a schematic diagram to summarize the experimental outline of the paper.
  3. Missing description on the generation of clonal cell lines in material and method.

Reviewer 3 Report

The major goal of transforming a transient signal to a constant signal stored in the genome is interesting. However, in principle, it does not need to turn to a relatively complicate synthetic notch signaling module. For example, doxcycline & doxcycline-inducible gene expression can easily achieve this aim. The authors need to highlight the necessity of using this system for certain applications. Meanwhile, the authors would also need to correlate and distinguish from a recent publication with similar concept (doi: 10.1007/s13238-020-00690-1).

The writing of the manuscript needs to be significantly improved.

(1) The abstract should provide a concise summary of the background, novelty, the main conclusion and the importance. The current abstract can be distracting with too much background information.

(2) The introduction contains a large amount of unrelated background, for example, TCR, CAR mechanisms, etc. There are a lot of background information not from this research but lack of citation.

(3) Figure legends are not written in the correct way. They should describe the data in the figures themself, neither how the experiments were carried out, nor the rationale of the experimental design.

Regarding the figures and experiments:

(1) In Fig. 1a-c, the authors attempt to select synNotch positive cells and rule out unspecific activation by cell sorting after the activation. However, the non-specific activation largely comes from the nature of synNotch receptor. This sorting would not contribute much to reduce the nonspecific activation, unless the author can provide evidence against this opinion. A typical method to sort out the positive cells is to have a constitutive reporter on the responsive construct.

(2) In Fig. 2, an important control is missing: sender cells with a nonspecific signal. This control tells how much nonspecific activation it leads to and whether the activation achieved in the current result is significant. Whenever statistics is done, it is needed to describe how, what n number means, and what is the P value.

(3) In Fig. 3, a similar control as described above is needed. The authors also need to show how many repeats and replicates were performed. And comparing with the above control, whether the result is significantly different after statistical analysis. A T7E1 enzyme digestion assay would be a good experiment to strengthen the genome cutting efficiency.